# Self-Adaptable Templates for Feature Coding

**Xavier Boix**[1,2*]    **Gemma Roig**[1,2*]    **Salomon Diether**[1]    **Luc Van Gool**[1]

[1]Computer Vision Laboratory, ETH Zurich, Switzerland
[2]LCSL, Massachusetts Institute of Technology & Istituto Italiano di Tecnologia, Cambridge, MA
{xboix,gemmar}@mit.edu
{boxavier,gemmar,sdiether,vangool}@vision.ee.ethz.ch

## Abstract

Hierarchical feed-forward networks have been successfully applied in object recognition. At each level of the hierarchy, features are extracted and encoded, followed by a pooling step. Within this processing pipeline, the common trend is to learn the feature coding templates, often referred as codebook entries, filters, or over-complete basis. Recently, an approach that apparently does not use templates has been shown to obtain very promising results. This is the second-order pooling (O2P) [1, 2, 3, 4, 5]. In this paper, we analyze O2P as a coding-pooling scheme. We find that at testing phase, O2P automatically adapts the feature coding templates to the input features, rather than using templates learned during the training phase. From this finding, we are able to bring common concepts of coding-pooling schemes to O2P, such as feature quantization. This allows for significant accuracy improvements of O2P in standard benchmarks of image classification, namely Caltech101 and VOC07.

## 1   Introduction

Many object recognition schemes, inspired from biological vision, are based on feed-forward hierarchical architectures, *e.g.* [6, 7, 8]. In each level in the hierarchy, the algorithms can be usually divided into the steps of feature coding and spatial pooling. The feature coding extracts similarities between the set of input features and a set of *templates* (the so called filters, over-complete basis or codebook), and then, the similarity responses are transformed using some non-linearities. Finally, the spatial pooling extracts one single vector from the set of transformed responses. The specific architecture of the network (*e.g.* how many layers), and the specific algorithms for the coding-pooling at each layer are usually set for a recognition task and dataset, *cf.* [9].

Second-order Pooling (O2P) is an alternative algorithm to the aforementioned coding-pooling scheme. O2P has been introduced in medical imaging to analyze magnetic resonance images [1, 2], and lately, O2P achieved state-of-the-art in some of the traditional computer vision tasks [3, 4, 5, 10]. A surprising fact of O2P is that it is formulated without feature coding templates [5]. This is in contrast to the common coding-pooling schemes, in which the templates are learned during a training phase, and at testing phase, the templates remain fixed to the learned values.

Motivated by the intriguing properties of O2P, in this paper we try to re-formulate O2P as a coding-pooling scheme. In doing so, we find that O2P actually computes similarities to feature coding templates as the rest of the coding-pooling schemes. Yet, what remains uncommon of O2P, is that the templates are "recomputed" for each specific input, rather than being fixed to learned values. In O2P, the templates are self-adapted to the input, and hence, they do not require learning.

From our formulation, we are able to bring common concepts of coding-pooling schemes to O2P, such as feature quantization. This allows us to achieve significant improvements of the accuracy

of O2P for image classification. We report experiments on two challenging benchmarks for image classification, namely Caltech101 [11], and VOC07 [12].

## 2   Preliminaries

In this Section, we introduce O2P as well as several coding-pooling schemes, and identify some common terminology in the literature. This will serve as a basis for the new formulation of O2P, that we introduce in the following section.

The algorithms that we analyze in this section are usually part of a layer of a hierarchical network for object recognition. The input to these algorithms is a *set* of feature vectors that come from the output of the previous layer, or from the raw image. Let $\{\mathbf{x}_i\}_N$ be the set of input feature vectors to the algorithm, which is the set of $N$ feature vectors, $\mathbf{x}_i \in \mathbb{R}^M$, indexed by $i \in \{1, \ldots, N\}$. The output of the algorithm is a single vector, which we denote as $\mathbf{y}$, and it may have a different dimensionality than the input vectors.

In the following subsections, we present the algorithms and terminology of template-based methods, and then, we introduce the formulation of O2P that appears in the literature that apparently does not use templates.

### 2.1   Coding-Pooling based on Evaluating Similarities to Templates

Template-based methods are build upon similarities between the input vectors and a set of *templates*. Depending on the terminology of each algorithm, the templates may be denoted as filters, codebook, or over-complete basis. From now on, we will refer to all of them as templates. We denote the set of templates as $\{\mathbf{b}_k \in \mathbf{R}^M\}_P$. In this paper, $\mathbf{b}_k$ and the input feature vectors $\mathbf{x}_i$ have the same dimensionality, $M$. The set of templates is fixed to learned values during the training phase. There are many possible learning algorithms, but analyzing them is not necessary here.

The algorithms that are interesting for our purposes, start by computing a similarity measure between the input feature vectors $\{\mathbf{x}_i\}_N$ and the templates $\{\mathbf{b}_k\}_P$. Let $\Gamma(\mathbf{x}_i, \mathbf{b}_k)$ be the similarity function, which depends on each algorithm. We define $\boldsymbol{\gamma}_i$ as the vector that contains the similarities of $\mathbf{x}_i$ to the set of templates $\{\mathbf{b}_k\}$, and $\boldsymbol{\gamma} \in \mathbf{R}^{M \times P}$ the matrix whose columns are the vectors $\boldsymbol{\gamma}_i$, *i.e.*

$$\gamma_{ki} = \Gamma(\mathbf{x}_i, \mathbf{b}_k). \tag{1}$$

Once $\boldsymbol{\gamma}$ is computed, the algorithms that we analyze apply some non-linear transformation to $\boldsymbol{\gamma}$, and then, the resulting responses are merged together, with the so called pooling operation. The pooling consists on generating one single response value for each template. We denote as $g_k(\boldsymbol{\gamma})$ the function that includes both the non-linear transformation and the pooling operation, where $g_k : \mathbb{R}^{M \times P} \to \mathbb{R}$. We include both operations in the same function, but in the literature it is usually presented as two separate steps. Finally, the output vector $\mathbf{y}$ is built using $\{g_k(\boldsymbol{\gamma})\}_P$, $\{\mathbf{b}_k\}_P$ and $\{\mathbf{x}_i\}_N$, depending on the algorithm. It is also quite common to concatenate the outputs of neighboring regions to generate the final output of the layer.

We now show how the presented terminology is applied to some methods based on evaluating similarities to templates, namely assignment-based methods and Fisher Vector. In the sequel, these algorithms will be a basis to reformulate O2P.

**Assignment-based Methods**   The popular Bag-of-Words and some of its variants fall into this category, *e.g.* [13, 14, 15]. These methods consist on assigning each input vector $\mathbf{x}_i$ to a set of templates (the so called vector quantization), and then, building a histogram of the assignments, which corresponds to the average pooling operation.

We now present them using our terminology. After computing the similarities to the templates, $\boldsymbol{\gamma}$ (usually based on $\ell_2$ distance), $g_k(\boldsymbol{\gamma})$ computes both the vector quantization and the pooling. Let $s$ be the number of templates to which each input vector is assigned, and let $\boldsymbol{\gamma}'_i$ be the resulting assignment vector of $\mathbf{x}_i$ (*i.e.* $\boldsymbol{\gamma}'_i$ is the result of applying vector quantisation on $\mathbf{x}_i$). $\boldsymbol{\gamma}'_i$ has $s$ entries set to 1 and the rest to 0, that indicate the assignment. Finally, $g_k(\boldsymbol{\gamma})$ also computes the pooling for the assignments corresponding to the template $k$, *i.e.* $g_k(\boldsymbol{\gamma}) = \frac{1}{N} \sum_{i < N} \gamma'_{ki}$. The final output vector is the concatenation of the resulting pooling of the different templates, $\mathbf{y} = (g_1(\boldsymbol{\gamma}), \ldots, g_P(\boldsymbol{\gamma}))$.

**Fisher Vectors**    It uses the first and second order statistics of the similarities between the features and the templates [16]. Fisher Vector builds two vectors for each template $\mathbf{b}_k$, which are

$$\mathbf{\Phi}_k^{(1)} = \frac{1}{A_k} \sum_{i<N} \gamma_{ki} \left(\mathbf{b}_k - \mathbf{x}_i\right) \quad \mathbf{\Phi}_k^{(2)} = \frac{1}{B_k} \sum_{i<N} \gamma_{ki} \left((\mathbf{b}_k - \mathbf{x}_i)^2 - C_k\right), \tag{2}$$

$$\text{where } \gamma_{ki} = \frac{1}{Z_k} \exp\left[-\frac{1}{2}(\mathbf{x}_i - \mathbf{b}_k)^t \mathbf{D}_k(\mathbf{x}_i - \mathbf{b}_k)\right]. \tag{3}$$

$A_k$, $B_k$, $C_k$ are learned constants, $Z_k$ a normalization factor and $\mathbf{D}_k$ is a learned constant matrix of the model. Note that in Eq. (3), $\gamma_{ki}$ is a similarity between the feature vector $\mathbf{x}_i$ and the template $\mathbf{b}_k$. The final output vector is $\mathbf{y} = (\mathbf{\Phi}_1^{(1)}, \mathbf{\Phi}_1^{(2)} \dots, \mathbf{\Phi}_P^{(1)}, \mathbf{\Phi}_P^{(2)})$. For further details we refer the reader to [16].

We use our terminology to do a very simple re-write of the terms. We define $g_k(\boldsymbol{\gamma})$ and $\mathbf{b}_k^F$ (we use the super-index $F$ to indicate that are from Fisher vectors, and different from $\mathbf{b}_k$) as

$$g_k(\boldsymbol{\gamma}) = \|(\mathbf{\Phi}_k^{(1)}, \mathbf{\Phi}_k^{(2)})\|^2, \quad \mathbf{b}_k^F = \frac{1}{g_k(\boldsymbol{\gamma})}(\mathbf{\Phi}_k^{(1)}, \mathbf{\Phi}_k^{(2)}). \tag{4}$$

We can see the templates of Fisher vectors, $\mathbf{b}_k^F$, are obtained from computing some transformations to the original learned template $\mathbf{b}_k$, which involve the input set of features $\{\mathbf{x}_i\}$. $g_k(\boldsymbol{\gamma})$ is the norm of $(\mathbf{\Phi}_k^{(1)}, \mathbf{\Phi}_k^{(2)})$, which gives an idea of the importance of each template in $\{\mathbf{x}_i\}$, similarly to $g_k(\boldsymbol{\gamma})$ in assignment-based methods. Note that $\mathbf{b}_k^F$ and $g_k(\boldsymbol{\gamma})$ are related to only one fixed template, $\mathbf{b}_k$. The final output vector becomes $\mathbf{y} = (g_1(\boldsymbol{\gamma})\mathbf{b}_1^F, \dots, g_P(\boldsymbol{\gamma})\mathbf{b}_P^F)$.

## 2.2    Second-Order Pooling

Second-order Pooling (O2P) was introduced in medical imaging to describe the voxels produced in diffusion tensor imaging [1], and to process tensor fields [2, 17]. O2P starts by building a correlation matrix from the set of feature (column) vectors $\{\mathbf{x}_i \in \mathbb{R}^M\}_N$, *i.e.*

$$\mathbf{K} = \frac{1}{N} \sum_{i<N} \mathbf{x}_i \mathbf{x}_i^t, \tag{5}$$

where $\mathbf{x}_i^t$ is the transpose vector of $\mathbf{x}_i$, and $\mathbf{K} \in \mathbb{R}^{M \times M}$ is a square matrix. $\mathbf{K}$ is a symmetric positive definite (SPD) matrix, and contains second-order statistics of $\{\mathbf{x}_i\}$. The set of SPD matrices form a Riemannian manifold, and hence, the conventional operations in the Euclidean space can not be used. Several metrics have been proposed for SPD matrices, and the most celebrated is the Log-Euclidean metric [17]. Such metric consists of mapping the SPD matrices to the tangent space by using the logarithm of the matrix, $\log(\mathbf{K})$. In the tangent space, the standard Euclidean metrics can be used.

The logarithm of an SPD matrix can be computed in practice by applying the logarithm individually to each of the eigenvalues of $\mathbf{K}$ [18]. Thus, the final output vector for O2P can be written as

$$\mathbf{y} = vec\left(\log(\mathbf{K})\right) = vec\left(\sum_{k<M} \log(\lambda_k)\mathbf{e}_k \mathbf{e}_k^t\right), \tag{6}$$

where $\mathbf{e}_k$ are the eigenvectors of $\mathbf{K}$, and $\lambda_k$ the corresponding eigenvalues. The $vec(\cdot)$ operator vectorizes $\log(\mathbf{K})$.

In Eq. (6), apparently, there are no similarities to a set of templates. The absence of templates makes O2P look quite different from template-based methods. Recently, O2P achieved state-of-the-art results in some computer vision tasks, *e.g.* in object detection [3], semantic segmentation [5, 10], and for patch description [4]. Both reasons, motivates us to further analyze O2P in relation to template-based methods.

## 3    Self-Adaptability of the Templates

In this section, we introduce a formulation that relates O2P and template-based methods. The new formulation is based on comparing two final representation vectors, rather than defining how the

final vector $\mathbf{y}$ is built. We denote $\langle \mathbf{y}^r, \mathbf{y}^s \rangle$ as the inner product between $\mathbf{y}^r$ and $\mathbf{y}^s$, which are the final representation vectors from two sets of input feature vectors, $\{\mathbf{x}_i^r\}_N$ and $\{\mathbf{x}_i^s\}_N$, respectively, where we use the superscripts $r$ and $s$ to indicate the respective representation for each set. It will become clear during this section why we analyze $\langle \mathbf{y}^r, \mathbf{y}^s \rangle$ instead of $\mathbf{y}$.

We divide the analysis in three subsections. In subsection 3.1, we re-write the formulation of the template-based methods of Section 2 with the inner product $\langle \mathbf{y}^r, \mathbf{y}^s \rangle$. In subsection 3.2, we do the same for O2P, and this unveils that O2P is also based on evaluating similarities to templates. In subsection 3.3, we analyze the characteristics of the templates in O2P, which have the particularity that are self-adapted to the input.

### 3.1 Re-Formulation of Template-Based Methods

We re-write a generic formulation for the template-based methods described in Section 2 with the inner product between two final output vectors. The algorithms of Section 2 can be expressed as

$$\langle \mathbf{y}^r, \mathbf{y}^s \rangle = \sum_{k<P} \sum_{q<P} g_k(\boldsymbol{\gamma}^r) g_q(\boldsymbol{\gamma}^s) S(\mathbf{b}_k^r, \mathbf{b}_q^s), \qquad (7)$$

$$\text{where } \gamma_{ki} = \Gamma(\mathbf{x}_i, \mathbf{b}_k),$$

and $S(\mathbf{u}, \mathbf{v})$ is a similarity function between the templates that depends on each algorithm. Recall that $g_k(\boldsymbol{\gamma})$ is a function that includes the non-linearities and the pooling of the similarities between the input feature vectors and the the templates. To see how Eq. (7) arises naturally from the algorithms of Section 2, we now analyze them in terms of this formulation.

**Assignment-Based Methods** The inner product between two final output vectors can be written as

$$\langle \mathbf{y}^r, \mathbf{y}^s \rangle = (g_1(\boldsymbol{\gamma}^r), \dots, g_P(\boldsymbol{\gamma}^r))^t (g_1^s(\boldsymbol{\gamma}^s), \dots, g_P^s(\boldsymbol{\gamma}^s)) =$$

$$= \sum_{k<P} g_k(\boldsymbol{\gamma}^r) g_k(\boldsymbol{\gamma}^s) = \sum_{k<P} \sum_{q<P} g_k(\boldsymbol{\gamma}^r) g_q(\boldsymbol{\gamma}^s) \mathbf{I}(\mathbf{b}_k^r = \mathbf{b}_q^s), \qquad (8)$$

where the last step introduces an outer summation, and the indicator function $\mathbf{I}(\cdot)$ eliminates the unnecessary cross terms. Comparing this last equation to Eq. (7), we can identify that $S(\mathbf{b}_k^r, \mathbf{b}_q^s)$ is the indicator function (returns 1 when $\mathbf{b}_k^r = \mathbf{b}_q^s$, and 0 otherwise).

**Fisher Vectors** The inner product between two final Fisher Vectors is

$$\langle \mathbf{y}^r, \mathbf{y}^s \rangle = (g_1(\boldsymbol{\gamma}^r) \mathbf{b}_1^{rF}, \dots, g_P(\boldsymbol{\gamma}^r) \mathbf{b}_P^{rF})^t (g_1(\boldsymbol{\gamma}^s) \mathbf{b}_1^{sF}, \dots, g_P(\boldsymbol{\gamma}^s) \mathbf{b}_P^{sF})$$

$$= \sum_{k<P} \sum_{q<P} g_k(\boldsymbol{\gamma}^r) g_q(\boldsymbol{\gamma}^s) \mathbf{I}(\mathbf{b}_k^r = \mathbf{b}_q^s) \langle \mathbf{b}_k^{rF}, \mathbf{b}_q^{sF} \rangle. \qquad (9)$$

The indicator function appears for the same reason as in Assignment-Based Methods. The final templates for each set of input vectors, $\mathbf{b}_k^{rF}$, $\mathbf{b}_k^{sF}$, respectively, are compared with each other with the similarity $(\mathbf{b}_k^{rF})^t \mathbf{b}_k^{sF}$. Thus, $S(\mathbf{b}_k^{rF}, \mathbf{b}_q^{sF})$ in Eq. (7) is equal to $\mathbf{I}(\mathbf{b}_k^r = \mathbf{b}_q^s)(\mathbf{b}_k^{rF})^t \mathbf{b}_q^{sF}$.

### 3.2 O2P as Coding-Pooling based on Template Similarities

We now re-formulate O2P, in the same way as we did for template-based methods in the previous subsection. This will allow relating O2P to template-based methods, and show that O2P also uses similarities to templates.

We re-write the definition of O2P in Eq. (6) with $\langle \mathbf{y}^r, \mathbf{y}^s \rangle$. Using the property $vec(\mathbf{A})^t vec(\mathbf{B}) = tr(\mathbf{A}^t \mathbf{B})$, where $tr(\cdot)$ is the trace function of a matrix, $\langle \mathbf{y}^r, \mathbf{y}^s \rangle$ becomes (in the supplementary material we do the full derivation)

$$\langle \mathbf{y}^r, \mathbf{y}^s \rangle = \langle vec\left(\log(\mathbf{K}^r)\right), vec\left(\log(\mathbf{K}^s)\right) \rangle =$$

$$= \sum_{k<M} \sum_{q<M} \log(\lambda_k^r) \log(\lambda_q^s) \langle \mathbf{e}_k^r, \mathbf{e}_q^s \rangle^2, \qquad (10)$$

where $\mathbf{e}_k \mathbf{e}_k^t$ is a square matrix, and the eigenvectors, $\{\mathbf{e}_k^r\}_M$ and $\{\mathbf{e}_k^s\}_M$, are compared all against each other with $\langle \mathbf{e}_k^r, \mathbf{e}_q^s \rangle^2$. Going back to the generic formulation of template-based methods in

| Method | $S(\mathbf{b}_k^r, \mathbf{b}_q^s)$ | $\gamma_{ki} = \Gamma(\mathbf{x}_i, \mathbf{b}_k)$ | *templates* | $g_k(\boldsymbol{\gamma})$ |
|---|---|---|---|---|
| Assignment-based | $\mathbf{I}(\mathbf{b}_k^r = \mathbf{b}_q^s)$ | $\langle \mathbf{x}_i, \mathbf{b}_k \rangle$ | fixed | $\frac{1}{N}\sum_i \gamma'_{ki}$ |
| Fisher Vectors | $\mathbf{I}(\mathbf{b}_k^r = \mathbf{b}_q^s)\langle \mathbf{b}_k^{sF}, \mathbf{b}_P^{sF} \rangle$ | Eq. (3) | fixed/adapted | $\|(\boldsymbol{\Phi}_k^{(1)}, \boldsymbol{\Phi}_k^{(2)})\|^2$ |
| O2P | $\langle \mathbf{b}_k^r, \mathbf{b}_q^s \rangle^2$ | $\langle \mathbf{x}_i, \mathbf{b}_k \rangle^2$ | self-adapted | $\log\left(\frac{1}{N}\sum_i \gamma_{ki}\right)$ |

Table 1: Summary Table of the elements of our formulation for Assignment-based methods, Fisher Vectors and O2P.

Eq. (7), we can see that the similarity function between the templates, $S(\mathbf{e}_k^r, \mathbf{e}_q^s)$, can be identified in O2P as $\langle \mathbf{e}_k^r, \mathbf{e}_q^s \rangle^2$. Also, note that in O2P the sums go over $M$, which is the number of eigenvectors, and in Eq. (7), go over $P$, which is the number of templates. Finally, $g_k(\boldsymbol{\gamma})$ in Eq. (7) corresponds to $\log(\lambda_k)$ in O2P.

At this point, we have expressed O2P in a similar way as template-based methods. Yet, we still have to find the similarity between the input feature vectors and the templates. For that purpose, we use the definition of eigenvalues and eigenvectors, *i.e.* $\lambda_k \mathbf{e}_k = \mathbf{K}\mathbf{e}_k$, and also that $tr(\mathbf{e}_k \mathbf{e}_k^t) = 1$ (the eigenvectors are orthonormal). Then, we can derive the following equivalence: $\lambda_k = \lambda_k tr(\mathbf{e}_k \mathbf{e}_k^t) = tr(\mathbf{K}\mathbf{e}_k \mathbf{e}_k^t)$. Replacing $\mathbf{K}$ by $\frac{1}{N}\sum_i \mathbf{x}_i \mathbf{x}_i^t$, we find that the eigenvalues, $\lambda_k$, can be written using the similarity between the input vectors, $\mathbf{x}_i$, and the eigenvectors, $\mathbf{e}_k$:

$$\lambda_k = \frac{1}{N}\sum_i tr((\mathbf{x}_i \mathbf{x}_i^t)(\mathbf{e}_k \mathbf{e}_k^t)) = \frac{1}{N}\sum_i \langle \mathbf{x}_i, \mathbf{e}_k \rangle^2. \tag{11}$$

Finally, we can integrate all the above derivations in Eq. (10), and we obtain that

$$\langle \mathbf{y}^r, \mathbf{y}^s \rangle = \sum_{k<M}\sum_{q<M} g_k(\boldsymbol{\gamma}^r) g_q(\boldsymbol{\gamma}^s) \langle \mathbf{e}_k^r, \mathbf{e}_q^s \rangle^2, \tag{12}$$

$$\text{where } g_k(\boldsymbol{\gamma}) = \log(\lambda_k) = \log\left(\frac{1}{N}\sum_{i<N} \gamma_{ki}\right), \tag{13}$$

$$\text{and } \gamma_{ki} = \Gamma(\mathbf{x}_i, \mathbf{e}_k) = \langle \mathbf{x}_i, \mathbf{e}_k \rangle^2. \tag{14}$$

We can see by analyzing Eq. (12) that this equation takes the same form as the general equation of template-based methods in Eq. (7). Note that the eigenvectors take the same role as the set of templates, *i.e.* $\mathbf{b}_k = \mathbf{e}_k$ and $P = M$. Also, observe that $S(\mathbf{b}_k^r, \mathbf{b}_q^s)$ is the square of the inner product between eigenvectors, $\Gamma(\mathbf{x}_i, \mathbf{b}_k)$ is the square of the inner product between the input vectors and the eigenvectors, and the pooling operation is the logarithm of the average of the similarities. In Table 1 we summarize the corresponding elements of all the described methods.

### 3.3 Self-Adaptative Templates

We define self-adaptative templates as templates that only depend on the input set of feature vectors, and are not fixed to predefined values. This is the case in O2P, because the templates in O2P correspond to the eigenvectors computed from the set of input feature vectors. The templates in O2P are not fixed to values learned during the training phase. Interestingly, the final templates in Fisher Vectors, $\mathbf{b}_k^F$, are also partially self-adapted to the input vectors. Note that $\mathbf{b}_k^F$ are obtained by modifying the fixed learned templates, $\mathbf{b}_k$, with the input feature vectors.

Finally, note that in O2P the number of templates is equal to the dimensionality of the input feature vectors. Thus, in O2P the number of templates can not be increased without changing the input vectors' length, $M$. This begs the following question: do $M$ templates allow for sufficient generalization for object recognition for any set of input vectors? We analyze this question in the next section.

## 4 Application: Quantization for O2P

We observe in the experiments section that the performance of O2P degrades when the number of vectors in the set of input features increases. It is reasonable that $M$ templates are not sufficient when the number of different vectors in $\{\mathbf{x}_i\}_N$ increases, specially when they are very different

---
**Algorithm 1:** Sparse Quantization in O2P

---
**Input**: $\{\mathbf{x}_i\}_N$, $k$
**Output**: $\mathbf{y}$
**foreach** $i = \{1, \ldots, N\}$ **do**
$\quad\mid\quad \hat{\mathbf{x}}_i \leftarrow$ Set $k$ highest values of $\mathbf{x}_i$ to its vector entry: $x_i$, and the rest to 0
**end**
$\mathbf{K} = \frac{1}{N} \sum_i \hat{\mathbf{x}}_i \hat{\mathbf{x}}_i^t$
$\mathbf{y} = vec(\log(\mathbf{K}))$

---

from each other. We now introduce an algorithm to increase the robustness of O2P to the variability of the input vectors.

We quantize the input feature vectors, $\{\mathbf{x}_i\}$, before computing O2P. Quantization may discard details, and hence, reduce the variability among vectors. In the experiments section it is reported that this allows preventing the degradation of performance in object recognition, when the number of input feature vectors increases. The quantization algorithm that we use is sparse quantization (SQ) [15, 19], because SQ does not change the dimensionality of the feature vector. Also, SQ is fast to compute, and does not increase the computational cost of O2P.

**Sparse Quantization for O2P**    For the quantization of $\{\mathbf{x}_i\}$ we use SQ, which is a quantization to the set of $k$-sparse vectors. Let $\mathbb{R}_k^q$ be the set of $k$-sparse vectors, *i.e.* $\{\mathbf{s} \in \mathbb{R}^q : \|\mathbf{s}\|_0 \leq k\}$. Also, we define $\mathbb{B}_k^q = \{0,1\}_k^q = \{\mathbf{s} \in \{0,1\}^q : \|\mathbf{s}\|_0 = k\}$, which is the set of binary vectors with $k$ elements set to one and $(q - k)$ set to zero. The cardinality of $|\mathbb{B}_k^q|$ is equal to $\binom{q}{k}$. The quantization of a vector $\mathbf{v} \in \mathbb{R}^q$ into a codebook $\{\mathbf{c}_i\}$ is a mapping of $\mathbf{v}$ to the closest element in $\{\mathbf{c}_i\}$, *i.e.* $\hat{\mathbf{v}}^\star = \arg\min_{\hat{\mathbf{v}} \in \{\mathbf{c}_i\}} \|\hat{\mathbf{v}} - \mathbf{v}\|^2$, where $\hat{\mathbf{v}}^\star$ is the quantized vector $\mathbf{v}$. In the case of SQ, the codebook $\{\mathbf{c}_i\}$ contains the set of $k$-sparse vectors. These may be any of the previously introduced types: $\mathbb{R}_k^q, \mathbb{B}_k^q$. An important advantage of SQ over a general quantization is that it can be computed much more efficiently. The naive way to compute a general quantization is to evaluate the nearest neighbor of $\mathbf{v}$ in $\{\mathbf{c}_i\}$, which may be costly to compute for large codebooks and high-dimensional $\mathbf{v}$. In contrast, SQ can be computed by selecting the $k$ higher values of the set $\{v_i\}$, *i.e.* for SQ into $\mathbb{R}_k^q$, $\hat{v}_i = v_i$ if $i$ is one of the k-highest entries of vector $\mathbf{v}$, and 0 otherwise. For SQ into $\mathbb{B}_k^q$, the dimension indexed by the k-highest are set to 1 instead of $v_i$, and 0 otherwise. (We refer the reader to [15, 19] for a more detailed explanation on SQ).

In Algorithm 1 we depict the implementation of SQ in O2P, which highlights its simplicity. The computational cost of SQ is negligible compared to the cost of computing O2P. We use the set of $k$-sparse vectors in $\mathbb{R}_k^M$ for SQ, which worked best in practice, as shown in the following.

## 5    Experiments

In this section, we analyze O2P in image classification from dense sampled SIFT descriptors. This setup is common in image classification, and it allows direct comparison to previous works on O2P. We report results on the Caltech101 [11] and VOC07 [12] datasets, using the standard evaluation benchmarks, which are the mean average precision accuracy across all classes.

### 5.1    Implementation Details

We use the standard pipeline for image classification. We never use flipped or blurred images to extend the training set.

**Pipeline.**    For Caltech101, the image is re-sized to take a maximum height and width of 300 pixels, which is the standard resizing protocol for this dataset. For VOC07 the size of the images remains the same as the original. We extract SIFT [8] from patches on a regular grid, at different scales. In Caltech 101, we extract them at every 8 pixels and at the scales of 16, 32 and 48 pixels diameter. In VOC07, SIFT is sampled at each 4 pixels and at the scales of 12, 24 and 36 pixels diameter. O2P is computed using the SIFT descriptors as input, and using spatial pyramids. In

Caltech101, we generate the pooling regions dividing the image in $4 \times 4$, $2 \times 2$ and $1 \times 1$ regions, and in VOC07 in $3 \times 1$, $2 \times 2$ and $1 \times 1$ regions. To generate the final descriptor for the whole image, we concatenate the descriptors for each pooled region. We apply the power normalization to the final feature dimensions, $sign(x)|x|^{3/4}$, that was shown to work well in practice [5]. Finally, we use a linear one-versus-rest SVM classifier for each class with the parameter $C$ of the SVM set to 1000. We use the LIBLINEAR library for the SVM[20].

**Other Feature Codings.** As a sanity check of our results, we replace O2P with the Bag-of-Words [13] baseline, without changing any of the parameters. In Caltech101, we replace the average pooling of Bag-of-Words by max-pooling (without normalization) as it performs better. The codebook is learned by randomly picking a set of patches as codebook entries, which was shown to work well for the encodings we are evaluating [14]. We use a codebook of 8192 entries, since with more entries the performance does not increase significantly, but the computational cost does.

## 5.2 Results on Caltech101

We use 3 random splits of 30 images per class for training and the rest for testing. In Fig. 1a, results are shown for different spatial pyramid configurations, as well as different levels of quantization. Note that SQ with $k = 128$ is not introducing any quantization, as SIFT features are 128 dimensional vectors. Note that using SQ increases the performance more than 5% compared to when not using SQ ($k = 128$), when using only the first level of the pyramid. For the other levels of the pyramid, there is less improvement with SQ. This is in accordance with the observation that in smaller regions there are less SIFT vectors, the variability is smaller, and the limited amount of templates is able to better capture the meaningful information than in bigger regions. We can also see that for small $k$ of SQ, the performance degrades due to the introduction of too much quantization.

We also run experiments with Bag-of-Words with max-pooling (74.8%), and O2P without SQ (76.52%), and both of them are surpassed by O2P with SQ (78.63%). In [5], O2P accuracy is reported to be 79.2% with SIFT descriptor (we do not compare to their version of enriched SIFT, since all our experiments are with normal SIFT). We inspected the code of [5], and we found that the difference of accuracy mainly comes from using a more drastic resizing of the image, that takes a maximum of 100 pixels of width and height (usually in the literature it is 300 pixels). Note that resizing is another way of discarding information, and hence, O2P may benefit from that. We confirm this by resizing the image back to 300 pixels in [5]'s code, and the accuracy is 77.1%, similar to the one that we report without SQ in our code. The accuracy is not exactly the same due to differences in the SIFT parameters in [5]. Also, we tested SQ in [5]'s code with the resizing to a maximum of 100 pixels, and the accuracy increased to 79.45%, which is higher than reported in [5], and close to state-of-the-art results using SIFT descriptors (80.3%) [21].

## 5.3 Results on VOC07

In Fig. 1b, we run the same experiment as in Caltech101. Note that the impact of SQ is even more evident than in Caltech101. In Table 2 we report the per-class accuracy, in addition to the mean average precision reported in Fig. 1b. We follow the evaluation procedure as described in [12].

With the full pyramid, when we use SQ the accuracy increases from 18.81% to 50.97%. In contrast to Caltech101, O2P with SQ performance is similar to our implementation of Bag-of-Words (51.14%). Thus, under adverse conditions for O2P, *i.e.* images with high variability such as in VOC07 and with a high number of input vectors, we can use SQ and obtain huge improvements of the O2P's accuracy. The best reported results [22] in VOC07 are around 10% better than O2P with SQ, yet we obtain more than 30% improvement from the baseline.

## 6 Conclusions

We found that O2P can be posed as a coding-pooling scheme based on evaluating similarities to templates. The templates of O2P self-adapt to the input, while the rest of the analyzed methods do not. In practice, our formulation was used to improve the performance of O2P in image classification. We are currently analyzing self-adaptative templates in deep hierarchical networks.

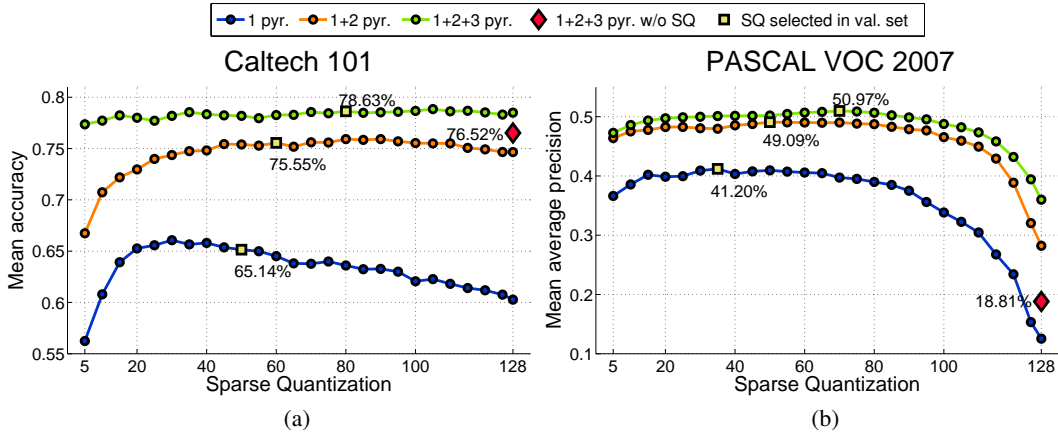

Figure 1: *Results for different numbers of non-zero entries of SQ.* Note that SQ at $k = 128$ is not introducing any quantization, since SIFT features are $128$ dimensional vectors. (a) Caltech 101 (using 30 images per class for training), (b) VOC07.

| | Aeroplane | Bicycle | Bird | Boat | Bottle | Bus | Car | Cat | Chair | Cow | Dinning Table | Dog | Horse | Motorbike | Person | Potted Plant | Sheep | Sofa | Train | TV/Monitor | Average |
|---|---|---|---|---|---|---|---|---|---|---|---|---|---|---|---|---|---|---|---|---|---|
| 3 Pyr. O2P + SQ | 72 | 53 | 45 | 63 | 23 | 51 | 69 | 52 | 50 | 35 | 44 | 41 | 74 | 56 | 78 | 19 | 35 | 50 | 67 | 45 | **50.97** |
| 3 Pyr. O2P w/o SQ | 34 | 9 | 12 | 18 | 6 | 19 | 40 | 14 | 26 | 14 | 9 | 21 | 28 | 17 | 55 | 7 | 7 | 10 | 16 | 12 | 18.81 |
| 2 Pyr. O2P + SQ | 71 | 50 | 41 | 62 | 20 | 50 | 68 | 47 | 47 | 33 | 41 | 37 | 69 | 56 | 74 | 18 | 36 | 51 | 66 | 44 | **49.09** |
| 1 Pyr. O2P + SQ | 66 | 41 | 32 | 58 | 15 | 37 | 58 | 38 | 40 | 27 | 28 | 30 | 61 | 43 | 66 | 20 | 33 | 37 | 56 | 36 | **41.20** |
| 1 Pyr. O2P w/o SQ | 21 | 7 | 11 | 9 | 6 | 8 | 29 | 10 | 22 | 4 | 7 | 12 | 12 | 8 | 49 | 6 | 5 | 7 | 9 | 9 | 12.53 |

Table 2: *PASCAL VOC 2007 classification results.* The average score provides the per-class average. We report results for O2P, with and without SQ, with the first plus second plus third levels of pyramids (3 Pyr.), O2P with SQ with the first plus second levels of pyramids (2 Pyr.), and O2P with and without SQ only with the first level of pyramids (1 Pyr.).

**Acknowledgments:** We thank the ERC for support from AdG VarCity.

## Footnotes

*Both first authors contributed equally.

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
