[Supplementary Material]

# Self-Adaptable Templates for Feature Coding

Xavier Boix[1,2*]   Gemma Roig[1,2*]   Salomon Diether[1]   Luc Van Gool[1]

[1]Computer Vision Laboratory, ETH Zurich, Switzerland

[2]LCSL, Massachusetts Institute of Technology & Istituto Italiano di Tecnologia, Cambridge, MA

{xboix,gemmar}@mit.edu

{boxavier,gemmar,sdiether,vangool}@vision.ee.ethz.ch

In the following Supplementary Material, we introduce the additional information and derivations that we promised in the main paper.

## 1  TR as Coding-Pooling based on Pattern Similarities

### 1.1  Derivation of $\langle \mathbf{y}^r, \mathbf{y}^s \rangle$ in TR

Recall that the final output vector in TR is $\mathbf{y} = vec\left(\log(\mathbf{K})\right) = vec\left(\sum_{k<M} \log(\lambda_k)\mathbf{e}_k\mathbf{e}_k^t\right)$, where $\mathbf{e}_k$ are the eigenvectors of $\mathbf{K}$, and $\lambda_k$ the corresponding eigenvalues. $\mathbf{K}$ is the correlation matrix of the input set of vectors, $\mathbf{K} = \frac{1}{N}\sum_{i<N}\mathbf{x}_i\mathbf{x}_i^t$. The $vec(\cdot)$ operator vectorizes $\log(\mathbf{K})$, and $\mathbf{x}_i^t$ is the transpose of vector $\mathbf{x}_i$.

Also, recall that we denote $\langle \mathbf{y}^r, \mathbf{y}^s \rangle$ as the inner product between $\mathbf{y}^r$ and $\mathbf{y}^s$, which are the final representation vectors from two sets of input feature vectors, $\{\mathbf{x}_i^r\}_N$ and $\{\mathbf{x}_i^s\}_N$, respectively, where we use the superscripts $r$ and $s$ to indicate the respective representation for each set.

We re-write the definition of TR with $\langle \mathbf{y}^r, \mathbf{y}^s \rangle$. $\langle \mathbf{y}^r, \mathbf{y}^s \rangle$ becomes

$$\langle \mathbf{y}^r, \mathbf{y}^s \rangle = \sum_{k<M}\sum_{q<M} \log(\lambda_k^r)\log(\lambda_q^s) tr\left((\mathbf{e}_k^r\mathbf{e}_k^{rt})(\mathbf{e}_q^s\mathbf{e}_q^{st})\right). \tag{1}$$

The derivation of the above equation is

$$\langle \mathbf{y}^r, \mathbf{y}^s \rangle = \tag{2}$$

$$vec\left(\sum_{k<M} \log(\lambda_k^r)\mathbf{e}_k^r\mathbf{e}_k^{rt}\right)^t vec\left(\sum_{q<M} \log(\lambda_q^s)\mathbf{e}_q^s\mathbf{e}_q^{st}\right) = \tag{3}$$

$$tr\left(\sum_{k<M} \log(\lambda_k^r)(\mathbf{e}_k^r\mathbf{e}_k^{rt})\sum_{q<M} \log(\lambda_q^s)(\mathbf{e}_q^s\mathbf{e}_q^{st})\right) = \tag{4}$$

$$\sum_{k<M}\sum_{q<M} \log(\lambda_k^r)\log(\lambda_q^s) tr\left((\mathbf{e}_k^r\mathbf{e}_k^{rt})(\mathbf{e}_q^s\mathbf{e}_q^{st})\right), \tag{5}$$

where the last equation is the same as Eq. (1). In the derivation from Eq. (3) to Eq. (4) we use the property $vec(\mathbf{A})^t vec(\mathbf{B}) = tr(\mathbf{A}^t\mathbf{B})$, where $tr(\cdot)$ is the trace function of a matrix. From Eq. (4) to Eq. (5) we reorder the terms, and take out of the trace function the $\log(\lambda_k)$, which is a scalar.

Finally, we rewrite Eq. (5) using the following equivalence $tr((\mathbf{u}\mathbf{u}^t)(\mathbf{v}\mathbf{v}^t)) = (\mathbf{u}^t\mathbf{v})^2 = \langle \mathbf{u}, \mathbf{v} \rangle^2$, and it becomes:

$$\langle \mathbf{y}^r, \mathbf{y}^s \rangle = \sum_{k<M}\sum_{q<M} \log(\lambda_k^r)\log(\lambda_q^s)\langle \mathbf{e}_k^r, \mathbf{e}_q^s \rangle^2. \tag{6}$$

## Footnotes

*Both first authors contributed equally.