[Reviews · NeurIPS 2014]

Submitted by Assigned_Reviewer_4

This paper re-forulates tensor representations as a coding-pooling scheme. The proposed approach achieves state-of-the-art results on two image classification tasks: Caltech101 and VOC07.

This paper is well written. Background of feature coding-pooling, and tensor representations is well introduced. The proposed method takes advantage of the merits of both techniques. Its effectiveness is confirmed by experimental results on Caltech101 and PASCAL VOC07.

One problem of the proposed method is, the two datasets used in the experiments, Caltech101 and PASCAL VOC07, are quite out-dated in the community of computer vision. It would be ideal to see results on more challenging datasets, such as Caltech 256 or PASCAL datasets of later years. Also, this paper uses classification accuracy on PASCAL VOC07, while the standard metric on this dataset is mean average precision. Reporting mean average precision would be helpful to compare the proposed method to existing approaches.
Summary: This paper proposes an image classification approach that takes advantage of both the coding-pooling scheme and tensor representation. The dataset used for performance evaluation is out-dated, but is enough to demonstrate the effectiveness of the proposed method.

Submitted by Assigned_Reviewer_42

The paper aims to provide a more general interpretation of a recent class of `tensor representations’ based on second order information. In particular the authors provide derivations in support of the claim that such second order pooling methodologies can be viewed as coding-pooling schemes that adapt the feature coding pattern to the input features. Moreover, the authors show that by incorporating coding into the new interpretation some improvements can be achieved in benchmarks (N.B. the results on VOC07 are significant; however, the results on Caltech101 by no means represent a significant improvement over the state of the art).

In general this is an interesting, clearly written paper (modulo some issues, having to do with the presentation of the contributions, described below) which provides valuable insight into a class of successful, emerging techniques. Overall I find this paper to be relevant and worth publishing at NIPS. I would appreciate if the authors considered the following comments.

- Comments

I would say that some of the main contributions associated with the (second-order) tensor representation are probably more attributable to [1],[16], than to [1],[2] (line 133). Equally so, line 142 should read `…celebrated Log-Euclidean metric [4,5]’ instead of just [4] as these have been introduced in the vision context simultaneously at ECCV12. Overall I would make it clear also that [3] uses a different metric than [4,5], for example. That’s why the approach authors take in baptizing slightly different methods as TR could be misleading in places like Table 1 or 2. Moreover, as far as I can understand the various approaches denoted Pyr TR w/o SQ = O2P [5]. Particularly as some of the code in [5] appears to be used (do the authors use the enriched SIFT features of [5] in their experiments with Caltech 101, or in the coded features they propose?), it would be appropriate to not `remove all traces’ and present existing methods as if they were proposed by the authors, under a new name. Best would be to conserve the original names and use new ones only when new elements are proposed by the authors, e.g. Pyr. Tr + SQ.

The list of references is good. The authors can consider adding:

The Devil Is in the Details: an Evaluation of Recent Feature Encoding Methods: Chatffiled et al., BMVC 2011.
Efficient Match Kernels between Sets of Features for Visual recognition, Bo et al., NIPS 2009.
which also discuss different encoding strategies (former) and provide potentially useful kernel interpretations for bag of words models (latter).
Summary: A more general interpretation of a recent class of `tensor representations’ based on second order information with some new feature description propositions based on coding. Relevant analysis and promising results. The relation to the state of the art should be made clearer and the original method names conserved for existing models where nothing new is added (pyramid + second-order-averaging + log-euclidean + power normalization= O2P[5]).

Submitted by Assigned_Reviewer_43

The paper starts from the Tensor Reconstruction (used in medical imaging e.g. for diffusion tensor MRI) and derives a form of the dot-product between two tensors that can be viewed through a non-linear transformation plus pooling. Then they use this transform and a sparsification of the feature vector for image classification.

Here are the main issues with the paper:

- The definition of self adaptative patterns is ambiguous.

It is not clear what "not fixed to predefined values" means. If "invariant to scaling" is self-adaptative, then the claim that the TR are self-adaptative is wrong since if a matrix is scaled, its eigenvalues are also scaled.

- The paper shows that the TR can be seen as a non-linear transformation+pooling. However, results using this transformation lag behind the state of the art and it is not clear how flexible this representation could be in obtaining better results.

Summary: The work has some merit and is interesting, but the paper lacks clarity and the experiments is not very convincing.
Author Feedback
Author rebuttal: We thank all the reviewers for their valuable comments. Below we address their concerns.
R4:
-out-dated datasets: It is true that it is always desirable a more extensive evaluation, yet, our experiments show the practical advantages of the re-formulation of TR. Also, results from most previous works are reported in the same version of the datasets that we use, which allows for direct comparison.
-classification accuracy on VOC07: we use the standard metric of this dataset, using the provided code in the benchmark.

R42:
We will address all the suggested comments:
-line 133 and 145: we will change it accordingly.
-mention different metric of [3]: we will do it.
-naming the methods: we do mention in the text that TR w\o SQ = O2P [5], but we will add it in the tables to make it clearer, as suggested.
-use of enriched SIFT of [5]: we do not use the enriched SIFT. We only use SIFT features in order to have a fair comparison to other methods. The result of the enriched SIFT is in the text. We will make it clearer in line 375.
-add references: We will do it.

R43:
-definition of self adaptative patterns: it seems there is a misunderstanding. Recall that in the paper we use patterns to denote filters, codebook or templates of feature encodings. Many methods learn the patterns at training phase, and at testing, the patterns are fixed to the learned values, and do not change. For instance, in a CNN, at training the filters are learned, and at testing the filters are fixed to the learned values, and used for the convolutions. In the paper we show that in TR the patterns are not learned at training phase. In TR, it is at testing, that the patterns are computed and adapted to the input. Note that this has nothing to do with invariance to scaling. All other reviewers agree that the paper is "clearly written" / "well written".
-Results using this transformation lag behind the state-of-the art: recently, Girshick et al. "Rich Feature Hierarchies for Accurate Object Detection and Semantic Segmentation" CVPR2014 reports that CNN and TR (O2P) achieve comparable results in semantic segmentation, see table 4 in that paper. This is an evidence that TR can be competitive with current state-of-the-art, and with the additional advantage of not requiring a learning of the patterns during training. We will include the reference in the discussion for completeness.